Electrochemical biosensors in healthcare services: bibliometric analysis and recent developments

Yunus Ghazala 1
http://orcid.org/0000-0002-4485-752X Singh Rachana 2
Raveendran Sindhu 3
http://orcid.org/0000-0001-5554-0598 Kuddus Mohammed 4 mkuddus@gmail.com
1 Department of Basic Science, University of Hail , Hail , Saudi Arabia
2 Amity Institute of Biotechnology, Amity University Uttar Pradesh , Lucknow, Uttar Pradesh , India
3 Department of Food Technology, TKM Institute of Technology , Kollam, Kerala , India
4 Department of Biochemistry, College of Medicine, University of Ha’il , Hail , Saudi Arabia
Phitsuwan Paripok
Electronic publication date: 2023 Jun 27
Publication date: 2023
Volume: 11
Electronic Location ID: e15566
Received 2023 Apr 6; Accepted 2023 May 24
Copyright: © 2023 Yunus et al.
Copyright year: 2023
Copyright holder: Yunus et al.
License: This is an open access article distributed under the terms of the Creative Commons Attribution License, which permits unrestricted use, distribution, reproduction and adaptation in any medium and for any purpose provided that it is properly attributed. For attribution, the original author(s), title, publication source (PeerJ) and either DOI or URL of the article must be cited.
License URL: https://creativecommons.org/licenses/by/4.0/

Keywords: Biosensor, Electrochemical biosensor, Multiplex assays, Healthcare, Bibliometric analysis, VOSviewer

Funding: The authors received no funding for this work.

==============================
Biosensors are nowadays being used in various fields including disease diagnosis and clinical analysis. The ability to detect biomolecules associated with disease is vital not only for accurate diagnosis of disease but also for drug discovery and development. Among the different types of biosensors, electrochemical biosensor is most widely used in clinical and health care services especially in multiplex assays due to its high susceptibility, low cost and small in size. This article includes comprehensive review of biosensors in medical field with special emphasis on electrochemical biosensors for multiplex assays and in healthcare services. Also, the publications on electrochemical biosensors are increasing rapidly; therefore, it is crucial to be aware of any latest developments or trends in this field of research. We used bibliometric analyses to summarize the progress of this research area. The study includes global publication counts on electrochemical biosensors for healthcare along with various bibliometric data analyses by VOSviewer software. The study also recognizes the top authors and journals in the related area, and determines proposal for monitoring research.

Introduction

There is an urgent demand to develop a sensitive, reliable, accurate, quick and handy analytical tool for clinical applications that can sense biomolecules (Akgonullu et al., 2020; Thaler & Luppa, 2019). Various conventional techniques such as chromatographic and spectroscopic, are available for quantitative measurement of biomolecules (Liu et al., 2019; Karami, Yaminia & Asl, 2020). Although these methods are precise but require exclusive and complex equipment and also demand trained and expert personnel to operate the equipment and sample preparation (Soler et al., 2020) and which makes it challenging to detect the biomolecules in real-time. The invention of Clark and Lyons in the 1960s to develop an analytical enzymatic device for the detection of glucose led to opening the door for research on biosensors (Clark & Lyons, 1962; Zucolotto, 2020) and since then it has seen tremendous advances. Researcher from all over the world has amalgamated various branch of science viz physics, chemistry, material science, biology, biotechnology, nanotechnology and electronics to develop powerful, automated and cost-effective diagnostic platforms (Artika, Wiyatno & Maroef, 2020; Mehta et al., 2020; Bhalla et al., 2020; Chakhalian et al., 2020; Qin et al., 2020; Khan, DeVoe & Andreescu, 2023) for the fast and real-time detection of biomolecules with highly specific sensing abilities (Samson, Navale & Dharne, 2020; Sumitha & Xavier, 2023; Wang et al., 2020; Wu et al., 2023). A biosensor is an analytical device that transforms biological molecules/reactions into the readable signal by biological substances. Biosensors consist of three components: first, the recognition element (a membrane with different biological structures); second, transducer; and third, electronic devices that amplify the signal and data presentation (Chen & Wang, 2020; Srivastava & Khare, 2021). The reactions are converted into a readable signal by two methods that include electrochemical transducer (Pacheco et al., 2018) and optical transducer (Erdem et al., 2019).

Sensitive and precise methods are needed for early diagnosis of diseases since it is essential for patient survival and successful prognosis. Electrochemical biosensor technologies offer the potential to assess health status at point of care, disease onset and progression, user-friendly, cost-efficient, and monitor treatment outcome through a non-invasive method. Electrochemical biosensors are particularly well suited for device integration across the various biosensing platforms because they may be easily minimized and integrated into an electronic acquisition module on a single chip (Wu et al., 2023). Recently, the applications of electrochemical biosensors in healthcare have expanded significantly (Khattak et al., 2021) that includes detection of foodborne pathogens (Yunus & Kuddus, 2020; Kulkarni, Ayachit & Aminabhavi, 2023), bacterial infections (Mehrannia et al., 2023) clinical diagnosis (Li et al., 2023a, 2023b), respiratory diseases (Gutierrez-Galvez et al., 2022; Lee, Bhardwaj & Jang, 2022; Deng et al., 2022; Li, Che & Deng, 2023; Liang et al., 2023); cancer (Zeng et al., 2022; Peng et al., 2022; Wang et al., 2023) etc.

The aim of this review article is to provide an updated and cumulative knowledge on research and development, along with publications, on electrochemical biosensors and their applications in healthcare services. This comprehensive review needed to fulfill the gap of knowledge between electrochemical biosensors and their application in healthcare sector; and will open the door for researchers and clinicians to develop novel tool and techniques to monitor on-the-spot biomolecules in clinical samples and will be beneficial for future research and development in the healthcare monitoring sector. The review will be useful for the scientists, academicians, clinicians, pathologist, biotechnologist, environmentalist and health care providers along with research scholars working in biomedical fields. It will also attract graduate and post-graduate students along with industry involve in research and development of biomedical tools.

The objectives and need of this review was prepared based on a comprehensive literature survey on the related topics (published and unpublished) by exploring various search engines/databases (Web of Science, Scopus, PubMed, Google Scholar) with the keywords related to biosensor. The actual search terms were “electrochemical biosensor” OR “electrochemical biosensors”; and “electrochemical biosensor” OR “electrochemical biosensors” AND Healthcare. Unrelated and superficial literature were screened and excluded during the preparation of review outline. The included self-structured objectives of the review were screened by two independent reviewers to resolve any biases or mistakes. The review included number of studies related to publication numbers, publications types, articles, authors, keywords, citations, and co-occurrence.

Biosensors, their types and applications

Biosensors are nowadays being used in various fields such as waste monitoring, health monitoring, diagnostics, biomedical testing, agricultural research, forensics, psychiatric diagnosis of patient management, etc. Traditional medical applications include health care monitoring, disease diagnosis and clinical analysis for biosensor medical clothing (Mowbray & Amiri, 2019; Kucherenko, Topolnikova & Soldatkin, 2019). Elaborating on the medical applications of biosensors, glucose biosensor has reported historical presence related to the regulation of diabetes. It is also used to rationalize glucose levels in the blood, a vital diabetes predictor (Bahl et al., 2020; Kamel & Khattab, 2020). The first oxygen biosensor was developed by Clark & Lyons (1962).

Updike & Hicks (1967) developed the first enzyme-based biosensor in 1967, developed on immobilization methods such as adsorption of enzymes, and ionic or covalent bonding. Frequently used enzymes for biosensors were oxidoreductases, peroxidases, amino oxidases and polyphenol oxidases (Wang, 2008; Akyilmaz, Yorganci & Asav, 2010; Venugopal, 2002). The microbial electrode (microbe-based sensor) was first reported by Diviès (1975). After that tissue-based sensors were developed using animal and plant sources by Rechnitz (1978) for confirmation of the amino acid arginine. The development of immunosensors was based on the affinity of antibodies with respective antigens. Wang (1998) established the DNA-based biosensor based on the ability of a single-stranded nucleic acid molecule to bind with its complementary strand. The binding takes place due to the formation of hydrogen bonds between the bases.

The magnetic biosensors were developed in 2014 by Scognamiglio et al. (2014) with high sensitivity and miniature size, used to detect magnetically and nanoparticles in microfluidic channels exploiting the magnetoresistance effect. Two types of piezoelectric biosensors are developed, the quartz crystal microbalance and surface acoustic wave device, that measures the changes in resonance frequency of a piezoelectric crystal as mass changes with the crystal structure. The optical transducer biosensors (Leatherbarrow & Edwards, 1999) can monitor light variation that occurs during the interaction of biological elements in the study (Tereshchenko et al., 2016). Another important feature of the biosensors is the processing speed using biomolecular detection. There are so many portable devices with biosensors, are available for point-of-care monitoring such as measuring glucose, pregnancy, addiction and many more (Tereshchenko et al., 2016; Moussilli, Falou & Shubair, 2018).

Biosensors benefits and advancement in the medical field

Though the application of biosensors in different fields of research developed at a slow pace, the advancement of biotechnology/nanotechnology and understanding of diseases has opened the pave in the diagnostics of diseases. The proficiency to detect biomolecules associated with the disease is vital not only for accurate diagnosis of disease but also crucial for biomedical research including drug discovery and development (Hosu et al., 2019). In recent years, the advancement in science (basic and clinical) has given the ability to identify biomarkers even at very low concentrations with the knowledge of prognostic roles in diagnosis and disease progression. The identification of extremely susceptible, accurate, fastidious, quantifiable and multiplexed biomarkers, depends on the potential diagnostic technology (Palchetti, 2016; Sharma et al., 2021). Biosensors are used in medical testing and research in order to fast and accurately identify disorders and provide significant amounts of health care. Electrochemical biosensors are utilized in cancer research to quickly and precisely diagnose biomarkers. These biosensors have metal-specific electrodes with the ability to spot hazardous metal concentrations in water. It can also detect threatening pathogens and biorecognition elements such as antibodies, enzymes, peptides, biomolecules etc. (Li, Stachowski & Zhang, 2015; Bahl et al., 2021; Omidfar et al., 2020).

Electrochemical biosensors for multiplex assays

Concurrent estimation and quantification of diverse materials from a single sample with enhanced reproducibility and reliability is the most demanding field for the achievement of effective and high throughput detection. Though there are entrenched immunological techniques with prevalent advantages over other screening diagnostic methods, one of the major challenges for immunoassay is accurate multiplexing (Anfossi et al., 2019). Precise, low-cost, and reliable immunosensors are vital for the early diagnosis and supervision of progressive diseases. The multiplexed sensor has come up as a promising approach for next-generation diagnostics (Gil Rosa et al., 2022). In the past decade, several strategies have been reported for multiplex sensors such as spatial multiplexing, barcode multiplexing etc. The chief transducer used in these multiplexing biosensors, to transfer the energy from the biorecognition process to a readable signal, includes electrochemical, electrical and optical methods. The electrochemical immunosensors are developed by combining the analytical power of electrochemical techniques and the target analyte-specific nature of antibodies on a solid-phase immunoassay (Contreras-Naranjo & Aguilar, 2019).

Electrochemical biosensors in healthcare services

The capabilities of the biosensors are best utilized in the manufacturing sector, especially in medical, healthcare and clinical services. Various potential applications of biosensors in healthcare are health monitoring, heart diagnosis, disease detection, retinal prostheses, contrast imaging in MRI and several others summarized in Fig. 1. This wide range of abilities has elevated the healthcare sector and given a boost to societal services (Tan et al., 2017; Kudr et al., 2021).

Figure 1 Potential applications of biosensor in healthcare services (adapted from Haleem et al., 2021).

Among all the types of biosensors, the electrochemical biosensor is the most widespread in clinical and healthcare services due to its high susceptibility, cost-effectiveness and small size (Palchetti & Mascini, 2008). The electrochemical biosensor employs electrochemical transduction that measures conductivity or electric resistance based on sensing methodology. It applies a chemical reaction containing fixed biomolecules and target analytes that affects calculated electric properties of solution such as electric current by the production of ions (Zhang et al., 2021). The potential applications of biosensors in the medical field are summarized in Table 1.

Table 1 Significant applications of biosensors in the medical field.

Applications	Descriptions	References	
Track biological abnormalities	Biosensors are used to monitor a patient’s vital signs and spot biological problems. Early detection and prevention save treatment costs. Correct identification and monitoring reduce healthcare costs. With the help of medical biosensors, proactive action could be taken leading to timely treatment. The advancement of technology could help to minimize the cost spent on hospital therapy.	Ahmed et al. (2014), Upadhyay & Verma (2015), Rodrigues et al. (2020), Guerrieri et al. (2019).	
Heart rate tracking	A unique invention of biosensors, made for continuous tracking of heart rates and transmitting physiological information, is available in form of smartwatches and athletic bands. These sensors are designed to detect the specific biomarkers that could help to understand the disease. The bioreceptor of the sensor receives the level and/or body function and stimulates related optical and electrochemical information. This technology is important for many biomedical applications and for industry 4.0 to accomplish several milestones.	Pindoo & Sinha (2020), Javaid & Haleem (2019), Zhang et al. (2021), Kowalczyk (2020), Patel et al. (2016).	
Track body chemistry	An implantable sensor is being developed which can be inserted under human skin and allows tracking of the patient’s body chemistry. This technology is been proven beneficial for diabetic patients’ monitoring and exercise habits.	Lin, Bariya & Javey (2020), Morales & Halpern (2018), Bahadır & Sezgintürk (2015).	
Diet monitoring	With Diet Sensor, which can also keep track of calorie supply, users may now examine the molecules in food. Anyone with a major food allergy understands that being overly cautious with food will result in significant problems since they have little control over food. An sophisticated nanosensor is used in medicine to analyze an inpatient’s breath for acetone molecules.	Liao & Schembre (2018), Hammond et al. (2016).	
Glucose monitoring	For the treatment of diabetic patients’ glucose monitoring by electrochemical test, strips are being popularly used. This wearable device uses biosensors for the continuous monitoring of glucose and was also developed for the diagnostics of maternity testing. This device is also ideal for cholesterol monitoring.	Un et al. (2021), Gifta & Rani (2020), Lee, Park & Seo, 2020.	
Management of disease	Now health care services are heavily dependent on biosensor-based applications for the detection and successful management of diseases. Quick and reliable diagnostics are expected from the point of care for modern medicine. The biosensors have already gained attention and applications in medical equipment like IoT devices, and wearable equipment and demand is increasing for low-cost, high accuracy and sensitivity equipment with their uses in various applications.	Jain et al. (2021), Ajami & Teimouri (2015)	
Detecting signs inpatient	Additionally, biosensors can help doctors identify patient symptoms even faster than they do using conventional methods. Biosensors are being used by health facilities in a variety of methods to control heart rate, blood sugar levels, and blood pressure. Real-time diagnostic testing for patients are also beneficial for doctors.	Li et al. (2021), Masson (2017).	
Tracking biological data	A biosensor-based medical patch is designed that can track biological data like body temperature and respiratory measurements. In recent years many kits have been developed that can be used by patients to monitor and track everyday health and also crucial disease like cancer.	Javaid et al. (2020), Kozitsina et al. (2018), Kerman et al. (2008).	
Tracking cell protein	The biosensor is designed to track live cell proteins in real time without using labels. It is based on the microfluidic platform with the capacity of label-free and multiplex detection.	Kuehn (2016), Rodríguez-Delgado et al. (2015), Manekiya & Donelli (2021).	
Biomolecular detection and measurement	This biosensor allows the monitoring of oxygen levels in the body systems in real time.	Carreiro et al. (2016), Strong et al. (2021), Veloso, Cheng & Kerman (2012).	
Treatment process tracking after surgery	This biosensor is very crucial for patient monitoring after surgery. It incorporates sensors that can manage implant infection and inflammation as well as a dissolved pressure sensor for the brain.	Xing & Yang (2014), Shetti et al. (2019).	
Tracking heart rhythms of a cardiac patient	This wearable biosensor device can track heart rhythms in cardiac patients, and the tension levels of employees and workers.	Tzianni et al. (2020), Perumal & Hashim (2014), Asal et al. (2018), Carpenter, Paulsen & Williams (2018).	

Diagnostic biosensors

Accurate diagnosis is the most significant factor for the identification of diseases (Rivet et al., 2011). An electrochemical medical biosensor was established by Chatterjee & Bandyopadhyay (2020) and was recommended to sense coronavirus-specific aptamer for binding with polymer matrix. The analytical characteristics showed that the biosensor is reproducible, specific, and stable and presented a molecular imprinted, polymer-based method for coronavirus detection (Chatterjee & Bandyopadhyay, 2020). Chen et al. (2019) reported an electrochemical medical biosensor for the detection of DNA methylation. This biosensor contains stem-loop-tetrahedron composite DNA probes that anchor at an Au nanoparticle-coated gold electrode, a restriction enzymes digestion of HpaII, signal amplification procedures incorporating electrodeposition of Au nanoparticles, hybridization chain reaction and HRP (horseradish peroxidase) enzyme catalysis. Under optimized conditions, they showed a wide range from 1 AM to 1 PM with a detection limit of 0.93 aM. They have also performed the recovery test and confirmed it as a potential platform for detecting DNA methylation during clinical incidents (Chen et al., 2019).

A significant population of the world suffering from heart failure. Although there are techniques for the detection of cardiovascular diseases such as enzyme-linked immunosorbent assay (ELISA), fluorometry, immunoaffinity column assay, etc. (Ooi et al., 2006; Caruso, Trunfio & Milazzo, 2010; Maurer, Burri & de Marchi, 2010; Caruso, Verde & Cabiati, 2012; Watson, Ledwidge & Phelan, 2011), these are very conscientious, demanding, labour and time intensive. The biosensor based on electric measurement uses biochemical molecular biorecognition for preferred selection with a specific biomarker of interest. Biosensors are also prevalently being used in the diagnosis of infectious diseases such as urinary tract infections. An encouraging biosensor is under development which could be used for anti-microbial susceptibility and identification of pathogens. Biosensors are crucial parts of analytical techniques for detecting and measuring biomolecular quantities. It enables researchers to keep an eye on the oxygen levels in systems in real time. It enables the body’s systems to more accurately replicate the actions of real organs. Small-scale bio-structures used in the development of novel medications are intended to reproduce certain organ functions, such as the absorption of oxygen from the air into the bloodstream (Carreiro et al., 2016; Strong et al., 2021; Veloso, Cheng & Kerman, 2012).

Biosensors in health monitoring

The use of glucose biosensors has shown tremendous clinical applications in diagnostics. The most widely used for diabetes mellitus is for anytime anywhere monitoring with accurate control over blood sugar levels (Scognamiglio, Pezzotti & Pezzotti, 2010). The biosensor for blood glucose monitoring secures its usage with 85% of colossal global market accounts (Rea, Polticelli & Antonacci, 2009). The measurement of interleukin-10 (IL-10) plays a very critical role in the identification of end-stage heart failure patients susceptible to adverse outcomes during the early phase of left ventricular assisted device implantation. A unique biosensor based on hafnium oxide was developed by Lee, Zine & Baraket (2012), for the detection of antigens. They have studied the interaction of recombinant human IL-10 with corresponding monoclonal antibodies for early cytokine detection after device implantation. The antigen-antibody interaction was determined by fluorescence patterns and electrochemical impedance spectroscopy and the bio-recognition of proteins was characterized (Lee, Zine & Baraket, 2012). A biosensor-based medical patch is designed that can track biological data like body temperature and respiratory measurements. In recent years, many kits have been developed that can be used by patients to monitor and track everyday health and also crucial diseases like cancer. Additionally, it may be used to assist doctors in confirming when their patients are due to take their prescribed medications. The device could become widely available in medical facilities obligations to the presence of biosensors in patients’ bodies and hospitals. Because they continue to participate in the personalization of healthcare, it can significantly enhance the patient experience (Javaid et al., 2020; Kozitsina et al., 2018; Kerman et al., 2008). Also, the biosensor is used to track live cell proteins with no labels in real time. This biosensor is based on a microfluidic system that consists of a cell module and module structures of biosensors and is located in a single channel in a zigzag manner. The crescendos of cell secretion are trailed by continuous tracking of spectral changes. This platform offers multiplex and label-free detection (Fig. 2) in form of a miniature chip-like instrument (Kuehn, 2016; Rodríguez-Delgado et al., 2015; Manekiya & Donelli, 2021).

Figure 2 Real-time detection of multiple analytes with multi-label and multi-electrode electrochemical biosensor.

By combining brain pressure sensors that dissolve in solution and sensors that regulate implant infection and inflammation, biosensors offer the chance to improve post-operative care. Mechanisms for precise and consistent processing, storing, and exchanging this information with the required stakeholders are essential if this data is to have a significant influence on patient care. Building public trust in the processing of patient information would depend on the efficient use of biosensors in medical treatment (Haleem et al., 2021; Xing & Yang, 2014; Shetti et al., 2019). Screen-printed electrodes keep wearable biosensor devices compact and deliver more specialized data, increasing the variety of potential applications. Cardiovascular patients’ heart rhythms can be immediately monitored. It can monitor the levels of stress experienced by service providers, police enforcement officials, miners, firemen, and more. Therefore, this technology may look for athletes’ fitness before and after physical training to maintain constant well-being and increase outcomes (Tzianni et al., 2020; Perumal & Hashim, 2014; Asal et al., 2018; Carpenter, Paulsen & Williams, 2018; Haleem et al., 2021).

Bibliometric analysis of electrochemical biosensors in healthcare

Bibliometric analyses are valuable in relating communication and providing information that may convey ideas and activate the research in the relevant field (Castells, 1996; Latour, 2005). Bibliometric study includes publication counts, citation analysis, keyword analysis and other evaluation (Melkers, 1993). We used modified method of Yunus (2020) for analysis (Yunus, 2020). Out of many existing software for bibliometric works, we used VOSviewer software by Van, Nic & Waltman (2009). Keywords are very useful and an important term to get research data in specific field. We used the Web of Science (core collection) data source that delivers authentic data with specific keywords. The data was extracted on 1st October 2022 with the keyword “electrochemical biosensor” OR “electrochemical biosensors” (All fields); and “electrochemical biosensor” OR “electrochemical biosensors” AND Healthcare (All fields). We conducted a number of studies related to publication numbers, publications types, articles, authors, keywords, citations, and co-occurrence; the findings are displayed under related sub-titles.

Publications trends on electrochemical biosensors for healthcare

The scientific publishing in all research area is increasing continuously with high rates (Larsen & Von Ins, 2010; Price & De, 1963). The quantity of articles reveals the most recent study that has been done on the topic. The numbers of publication on electrochemical biosensors used in healthcare are also growing constantly. Figure 3 compares publication counts on electrochemical biosensors and electrochemical biosensors used in healthcare for all timespan since 1979. The dataset suggests significant publications started from 2005 onwards and represents same trends in both ‘electrochemical biosensors’ and ‘electrochemical biosensors used in healthcare’. The total publication counts on the topic ‘electrochemical biosensors’ and ‘electrochemical biosensors used in healthcare’ are 8,331 and 5,620, respectively. The first article on electrochemical biosensor was published in 1979; however, the article describing use of electrochemical biosensors in healthcare was published in 1987.

Figure 3 Yearly number of publication on electrochemical biosensors and electrochemical biosensors for healthcare.

Publication types and funding trends

We included all types of research including journal articles, review articles, editorial, research support and many more from our foremost results and studied distinctly. The topmost publication type was journal articles with count 4,904 followed by review article (497), proceeding paper (280), meeting abstract (36), and early access (33) (Fig. S1). Regarding funding agency, which support research of electrochemical biosensors for health care, National Natural Science Foundation (China) is on the uppermost position along with 1,988 records fallowed by Fundamental Research Funds for the Central Universities, China (203), European Commission (166), National Basic Research Program of China (153) and Natural Science Foundation of Shandong Province (128). Figure 4 shows the top 10 research funding organizations worldwide.

Figure 4 Top 10 research funding organizations for electrochemical biosensor for healthcare.

Language of publications and countries

Publication language analysis revealed that the maximum number of publications are in English language that is 98.5%, followed by Chinese (1.223%), Czech (0.053%), and Korean (0.035%). There are only few number of publications in other languages. In the study of publication country, China is on the top with 47.41% out of 5,620 publications fallowed by USA (9.8%), India (6.77%), Iran (5.74%) and South Korea (4.96%). Top 20 countries with publication counts are reported in Fig. S2.

Journal titles

The Biosensors Bioelectronics includes maximum number of publications (714) which is 12.7% of total publications on electrochemical biosensor for health care, followed by Sensors and Actuators B Chemical (414) and Electroanalysis (192). Figure 5 indicates the uppermost 20 journals publishing the investigation associated to electrochemical biosensors for health care. The majority of publications come from the Biosensors and Bioelectronics journals, with relatively few coming from other journals that are consistent with Bradford’s law (Bradford, 1946).

Figure 5 Top 20 journals publishing electrochemical biosensor for health care.

Key author

The results of this study suggest that the inverse relationship between the number of publications and the number of writers is related to the Lotka law. At one side, many authors (10,104) published only one article and on the other hand, a few authors (58) had published over 10 articles (Fig. 6A). The leading author of a article is usually the one who has the most influence in the publication of the article. We find that Zhang Y is the top author in the dataset with 84 publications. Wang J is second highest with 76 followed by Yuan R with 65 publications (Fig. 6B). Figures 6A and 6B represent the inverse correlation.

Figure 6 (A) More authors with few publications. Many authors only contribute one or a small number of papers. (B) Few authors with more publications.

A small number of authors provide a large number of papers on the subject.

Utilizing the program VOSviewer, we performed co-authorship analysis to determine the links between the writers (www.vosviewer.com). Based on precise categorization, VOSviewer creates charts and images of the data (Van et al., 2010; Van, Nic & Waltman, 2011). In order to conduct the analysis, the documents with more than 25 authors were excluded and the lowest requirement for an author was three articles. Only 601 of the 18,700 writers had co-authorship relationships, according to analysis results, despite the overall number of authors. Figure 7 shows how the program generated and represented the 33 different groupings of co-authorship linkages. The groupings or clusters are distinguished using various colors. These clusters range in author count from minimum five (cluster 33) to maximum 55 (cluster 1) authors.

Figure 7 Co-authorship analyses with authors.

Co-authorship analyses with countries

The minimum and maximum numbers of documents for each country were set at three and 30, respectively, when analyzing co-authorship with countries using VOSviewer. Out of 100 nations, the results showed that only 70 have co-authorship links with 12 distinct groupings (Fig. S3). Different colors are used to distinguish between the groupings or clusters. These clusters range in size from two (cluster 12) to fourteen (cluster 1) different nations. Top three countries with maximum co-authorship with other countries include China, USA and South Korea. The co-authorship analyses give young researchers an advantage to collaborate with highly cited scientist in electrochemical biosensor research at international level.

Keyword analysis

All keywords, author keywords, and keywords plus are the three categories of keywords available in the WoS database. The research of all keyword is the focus of this study. The top 10 keywords along with their frequency and link strength are given in Table S1. The most used keyword is Electrochemical Biosensor, with a frequency of 2,472, followed by Biosensor (1,084), Sensor (909), Nanoparticles (787), Gold nanoparticles (651), DNA (535), Electrode (520), Immobilization (422), Graphene (391), and Biosensors (374). These important keywords assist new users in their search for relevant information or material on the website. Multiple keywords may be combined to find the specific literatures, providing exact information in line with the study’s goal. The researcher working on a DNA-based electrochemical biosensor, for instance, may utilize the keywords DNA and electrochemical biosensor.

Co-occurrence study of all keywords and author keywords

The research of keyword co-occurrences is an appropriate method for identifying the primary subjects of publications. Using the VOSviewer tool, we produced two different types of analyses, including “all keywords” and “author keyword,” to detect co-occurrences (Yunus, 2020). Data pre-processing and clean-up were done using the thesaurus option. The gadget designed the overall co-occurrence with other keywords after setting the lowest frequency of keywords during the research to “2”. There are 15,930 total keywords, and 4,746 of them cross the threshold. In order to use in the research and visualizations, we selected the top 1,000 keywords. The network visualization of all keywords is displayed in Fig. 8A. Nine clusters were formed from these terms. A total of 190 items are in cluster 1. Cluster 4 has the top keyword, Electrochemical biosensor, which has a total link strength of 18,072. Electrochemical biosensor, biosensor, sensor, nanoparticle, and gold nanoparticle are the top five keywords. It could automatically demonstrate how the terms are related. The frequency of keywords may be reflected in the size of nodes; the more frequently occurring a term, the greater the node size. According to Fig. 8A, the Electrochemical biosensor node is the largest, indicating that this category of keywords occurs the most frequently. Additionally, nanoparticles also occur more frequently.

Figure 8 (A) Co-occurrence study of all keywords. (B) Co-occurrence of author keywords. (C) Author keywords co-occurrence analysis with density visualization.

In contrast to all keywords, author keywords are the focus of this sector. The same approach described in the preceding section. The minimum was set at five occurrences. There are 10,056 total keywords, and 589 of them were found. With the highest overall bond power, we chose the top 589 terms. The authors’ primary keyword at this moment is “Electrochemical biosensor”. Sixteen clusters that were discovered through analysis are shown in Fig. 8B. It offers the data to identify the novel keywords, which writers develop through co-occurrence analysis. Figure 8C shows a co-occurrence study of the author’s keywords together with a density visualization. The term Electrochemical biosensor, which is the densest, is shown by color.

Citation analysis of documents

The minimal number of occurrences in the analysis of document citations was set at 1. Total 5,033 of the 5,608 mentioned articles were deemed sufficient. The greatest collection of linked items has 1,000 items in it. The number of citation links was determined for each of the 996 documents and was visualized. Figure 9 illustrates the 16 total clusters that were created. The first cluster has 124 elements, whereas cluster 16 has just 25 elements. Top 20 documents with their citations and links in decreasing order are presented in Fig. S4. The most influential document with maximum citation (1,146) was “Direct electrochemistry of glucose oxidase and biosensing for glucose based on graphene” by Shan et al. (2009). The article describes formation of novel electrochemical biosensor by using glucose oxidase enzyme that showed potential application for the fabrication of novel glucose biosensors. This hypothesis was further utilized by many researchers for preparation of specific electrochemical biosensor. Another important article was “Electrochemical biosensors: recommended definitions and classification” with 937 citations that described definition and classification of electrochemical biosensors (Thévenot et al., 2001).

Figure 9 Citation analyses of documents.

Colors indicate cluster of related terms.

Co-citation analyses of reference

The minimum number of referenced references for co-citation analysis was set at 20. A total of 519 references out of 155,325 cited sources met the criteria. The total strength of co-citation linkages with each of the 519 cited references was determined and represented in Fig. S5 for each of them. All items grouped into six clusters viz. cluster 1 (189 items), cluster 2 (123 items), cluster 3 (59 items), cluster 4 (59 items), cluster 5 (59 items) and cluster 6 (31 items). The reference with most frequent co-citation “Laviron E, 1979, J Electroanal Chem, v101, p19, doi 10.1016/s0022-0728(79)80075-3” is in cluster 1 with total link strength of 878. The topmost 20 references with the highest overall link strength were given in Table 2. These references are the key source of content and literature in the file of electrochemical biosensor and its applications.

Table 2 The top 20 references with the most link strength.

Rank	Cited reference	Citations	Total link strength	
1	laviron e, 1979, j electroanal chem, v101, p19, doi 10.1016/s0022-0728(79)80075-3	182	878	
2	bard a.j., 2001, electrochemical meth	167	688	
3	hummers ws, 1958, j am chem soc, v80, p1339, doi 10.1021/ja01539a017	158	866	
4	drummond tg, 2003, nat biotechnol, v21, p1192, doi 10.1038/nbt873	121	781	
5	ronkainen nj, 2010, chem soc rev, v39, p1747, doi 10.1039/b714449k	113	690	
6	wang j, 2008, chem rev, v108, p814, doi 10.1021/cr068123a	109	601	
7	novoselov ks, 2004, science, v306, p666, doi 10.1126/science.1102896	108	853	
8	grieshaber d, 2008, sensors-basel, v8, p1400, doi 10.3390/s8031400	107	542	
9	ellington ad, 1990, nature, v346, p818, doi 10.1038/346818a0	106	664	
10	steel ab, 1998, anal chem, v70, p4670, doi 10.1021/ac980037q	101	657	
11	clark lc, 1962, ann ny acad sci, v102, p29, doi 10.1111/j.1749-6632.1962.tb13623.x	100	728	
12	tuerk c, 1990, science, v249, p505, doi 10.1126/science.2200121	95	539	
13	kamin ra, 1980, anal chem, v52, p1198, doi 10.1021/ac50058a010	89	406	
14	wang j, 2006, biosens bioelectron, v21, p1887, doi 10.1016/j.bios.2005.10.027	89	440	
15	[anonymous], [no title captured]	83	280	
16	deng y, 2013, j biomed nanotechnol, v9, p318, doi 10.1166/jbn.2013.1487	80	756	
17	fan ch, 2003, p natl acad sci usa, v100, p9134, doi 10.1073/pnas.1633515100	80	536	
18	shao yy, 2010, electroanal, v22, p1027, doi 10.1002/elan.200900571	79	617	
19	geim ak, 2007, nat mater, v6, p183, doi 10.1038/nmat1849	77	610	
20	deng y, 2013, j biomed nanotechnol, v9, p1378, doi 10.1166/jbn.2013.1633	74	713	

Future prospective of electrochemical biosensors in the medical field

Rapid detection of pathogens and early diagnosis of diseases are major challenges for scientific community to adapt proper treatment strategies to the healthcare sector. The recent pandemic (COVID-19) had devastating impact on humans due to shortfalls in early diagnosis of this disease. Biosensor based devices provides a safer, economic and more effective methods of diagnosis than the invasive methods such as blood tests (Sun et al., 2022). A potential technique for the earlier detection of Alzheimer’s disease was recently developed by scientists using an electrochemical sensing platform based on a superwettable microdroplet array to detect various Alzheimer’s disease biomarkers. Because to its exceptional qualities, including a large specific surface, extraordinary electrical conductivity, and superior biocompatibility, this superwettable electrochemical sensing platform demonstrated excellent sensitivity and a low detection limit (Huang et al., 2022). Liang et al. (2022) developed miniGFPs for live cell imaging in bacterial culture under anaerobic conditions that can be serve as a multisensing platform for fluorescence biosensor development for in vitro and in-cell applications. Electrochemical methods are fast, accurate, and nondestructive tools for analyzing a wide range of targeted targets and have gradually increased and prioritized medical purposes. However, transformation of electrochemical biosensing results from speculative studies to clinical application remain faces various obstacles. The important challenges include its sensitivity, specificity, multiplex capability, integration of different functions into a single biochip, and in vivo sensing capability. The development of new ultrasensitive transducer technology with cutting-edge research and creating smart, sensitive, and reliable biosensors could be beneficial for its clinical applications.

Even though there are promising research and developments on electrochemical biosensors in the last decade, availability of commercial and effective devices in the real-world setting is still faraway.

Conclusion

Nowadays, electrochemical biosensors are being used in various medical fields including diagnostics, forensics, disease diagnosis, clinical analysis, health care monitoring and drug discovery and development. In immunological techniques, one of the major challenges for immunoassay is precise multiplexing for the early diagnosis and supervision of progressive diseases. The multiplexed electrochemical biosensors have come up with promising approach for next-generation diagnostics, which target analyte-specific nature of antibodies on a solid-phase immunoassay. The electrochemical biosensor is the most used biosensor in clinical and healthcare services due to its small size, cost-effectiveness and high susceptibility. The research on electrochemical biosensors and its possible applications in healthcare are increasing continuously. Recently the scientist fabricated an immunosensor to detect transferrin levels indirectly in cancer patients. Transferrin level have been associated with different disease conditions and are known to play a crucial role in various malignancies (Kaur et al., 2023). Electrochemical biosensors are also used for the detection of breast cancer biomarkers (Chiorcea-Paquim, 2023). In future, the electrochemical biosensors can provide a highly sensitive test and may be used to detect toxins in very low concentrations. Toxins pose a serious threat to human health and can be misused by terrorist or military. In order to choose the best treatments or countermeasures, early detection for outdoor measurement or point-of-care diagnostics is crucial. The bibliometric studies of publications on the subject of electrochemical biosensors for healthcare showed sound advancement in associated articles, with a faster rate of advancement in the last 5 years. Most of the publications are influenced by the authors from China fallowed by USA, and India. The leading journals for publishing articles are Biosensors Bioelectronics followed by Sensors & Actuators B Chemical, and Electroanalysis. Article has the top spot among the various publishing types, followed by Review articles. In terms of financing for research, the National Natural Science Foundation of China comes out on top, followed by the Fundamental Research Funds for The Central Universities of China and the European Commission. The analysis also showed that more authors had just one publication than there were few authors with several publications, in addition to the finest authors in the linked subject. These results fallow the bibliometric distribution law. The five most important “all keywords” in a co-occurrence analysis are electrochemical biosensor, biosensor, sensor, nanoparticles and gold nanoparticles. Also, in analysis of ‘author keywords’, top two keywords were same that is electrochemical biosensor and biosensor. The density visualization analysis also suggested that the keywords electrochemical biosensor is a most arising keyword. Additionally, the document’s citation analysis and co-citation of references show that the publication by authors based in China is the most referenced source. This analysis exposed growth of publications on electrochemical biosensor which will be beneficial for future research and development in the related field. We expect that this review will provide a valuable basis for the development of novel electrochemical biosensors and their application in healthcare.

Supplemental Information

Supplemental Information 1 Ten most popular terms, along with their frequency.

Click here for additional data file.

Supplemental Information 2 Publication types of electrochemical biosensor(s) for healthcare with counts Publication types of electrochemical biosensor(s) for healthcare with counts.

Click here for additional data file.

Supplemental Information 3 Best 20 countries by number of publications.

Click here for additional data file.

Supplemental Information 4 Co-authorship analyses with countries. Colors specify cluster of related terms.

Click here for additional data file.

Supplemental Information 5 Top 20 documents with the citations and links.

Click here for additional data file.

Supplemental Information 6 Co-citation analyses of reference. Colors indicate cluster of related terms.

Click here for additional data file.

Additional Information and Declarations

Competing Interests

Author Contributions

Data Availability

Mohammed Kuddus is an Academic Editor for PeerJ.

Ghazala Yunus conceived and designed the experiments, performed the experiments, analyzed the data, prepared figures and/or tables, and approved the final draft.

Rachana Singh performed the experiments, analyzed the data, prepared figures and/or tables, and approved the final draft.

Sindhu Raveendran conceived and designed the experiments, performed the experiments, authored or reviewed drafts of the article, and approved the final draft.

Mohammed Kuddus conceived and designed the experiments, analyzed the data, prepared figures and/or tables, authored or reviewed drafts of the article, and approved the final draft.

The following information was supplied regarding data availability:

The raw data is available in the Supplemental Table and Figures.

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
