# Peer review of "Electrochemical biosensors in healthcare services: bibliometric analysis and recent developments"

_PeerJ, doi:10.7717/peerj.15566_

## Round 0.1 · original submission · Major Revisions

It would be helpful to make the Introduction more clear on the importance of biosensors for cutting-edge research in healthcare services.
- Please include more recent articles that explore current trends and potential applications of technology.
- It would be helpful to have each section's content balanced and to have a thorough discussion.
-Verify English

Reviewer 1 ·

Basic reporting

The paper presents an interesting bibliometric analysis of electrochemical biosensors in healthcare. However, the manuscript requires a major revision to provide more context and depth in certain sections, as well as to address methodological concerns and improve the overall clarity of the paper.

1. The introduction could benefit from a more detailed explanation of the importance of electrochemical biosensors in healthcare and their potential applications.
2. It would be helpful to clarify the criteria used for selecting the articles included in the bibliometric analysis, ensuring a comprehensive and representative dataset.
3. In Section 6.6, consider providing more context on the significance of the co-authorship analysis, and its implications for international collaboration in electrochemical biosensor research.
4. Section 6.7 presents the top 10 keywords used in the research, but it might be helpful to further discuss the significance of these keywords and how they relate to the development of electrochemical biosensors.
5. For the co-occurrence study in Section 6.8, consider adding a brief explanation of the methodology and the significance of the findings to provide readers with a clearer understanding.
6. The citation analysis in Section 6.9 should include a brief discussion of the most influential papers and their contributions to the field of electrochemical biosensors in healthcare.
7. In Section 6.10, the co-citation analysis of references should be further elaborated, explaining the importance of the top-cited references and their relevance to the topic.
8. Section 7 could benefit from a more in-depth exploration of the challenges faced in translating electrochemical biosensing results from research studies to clinical applications.
9. The discussion on the future prospective of electrochemical biosensors in Section 7 should also address potential limitations and obstacles that may need to be overcome in order to realize their full potential in healthcare.
10. Consider providing specific examples of current or near-future electrochemical biosensor applications in healthcare, highlighting their potential impact on diagnostics, disease management, and patient care.

Experimental design

no comment

Validity of the findings

no comment

Additional comments

no comment

Reviewer 2 ·

Basic reporting

no comment

Experimental design

no comment

Validity of the findings

no comment

Additional comments

Biosensors are nowadays being used in various fields including disease diagnosis and clinical analysis. Among the different types of biosensors, electrochemical biosensor is most widely used in clinical and health care services especially in multiplex assays due to its high susceptibility, low cost and small size. This article is focused on electrochemical biosensors for multiplex assays and in healthcare services. The authors used bibliometric analyses to summarize the progress of this research area. This review work can be encouraged, so this manuscript can be recommended for publication. Nevertheless, before the formal acceptance, it is suggested the authors should improve this manuscript as follows:

(1) As the authors claimed, this review article includes both bibliometric analysis and recent developments. However, many references are several years, or even 10 years ago. It is difficult to get to know the recent developments by analyzing so many old literatures.
(2) It is suggested that more references should be included and analyzed. Usually, a high quality review paper may have reasonable number of references. It would better if more than 150 references could be included in this paper.
(3) It is suggested that the authors should make a more careful literature survey. Some well-known colleagues in the electrochemical biosensor field are not mentioned, and their publications have been included. For instance, the publications of Prof. Genxi Li at Nanjing University who is an Associate Editor of Biosensors and Bioelectronics, the leading publishing medium in the biosensor field, cannot be found in the reference list.
(4) The presentation should be also improved. For instance, “Biosensors are now a days being used……” should be “Biosensors are nowadays being used……”. “Among the different types of biosensor,” should be “Among the different types of biosensors, ”.

---

## Round 0.2 · accepted · Accept

The manuscript is much improved. It can be accepted for publication.